

# Computer vision-based algorithm for precise defect detection and classification in photovoltaic modules

Jian Guo

Department of Information Engineering, Xiamen Ocean Vocational College, Xiamen, Fujian, China

## ABSTRACT

In recent years, driven by advancements in the photovoltaic industry, solar power generation has emerged as a crucial energy source in China and the globe. A progressive annotation approach is employed to pinpoint and label defect samples to enhance the precision of automated detection technology for minor defects within photovoltaic modules. Subsequently, computer vision techniques are harnessed to segment photovoltaic modules and defect samples amidst intricate backgrounds accurately. Finally, a transfer learning training model is deployed to classify and identify defects effectively. The results indicate that the mask-region convolutional neural network model achieves remarkable accuracy and recall rates of 98.7% and 0.913, respectively. Furthermore, the detection speed and inference time are 280.69 frames per second and 3.53 ms, respectively. In essence, the defect detection and classification algorithm utilizing computer vision techniques significantly enhances the precision of automated detection technology in identifying minor defects within complex environments. This advancement holds profound practical significance in ensuring photovoltaic modules' quality and operational reliability.

## INTRODUCTION

With the rapid development of clean energy, solar energy has become the backbone of global renewable energy research (*Li et al., 2019*). However, photovoltaic (PV) systems are distributed outdoors in complex environments and variable climates. PV modules are inevitably damaged during transportation, installation, and use (*Su, Chen & Zhou, 2021*). At the same time, in complex outdoor environments, detecting hidden cracks on PV modules is prone to interference, which also brings difficulties to the defect detection of PV modules. Although the quality of PV modules directly affects the system's operational efficiency, expensive manual inspections and frequent failures of PV modules are prominent issues in the current operation and maintenance of PV systems (*Du et al., 2019*). With the advancement in computer technology, advanced computer vision technology has brought new ideas to automated detection technology. Computer vision technology with deep learning and image processing capabilities is skilled in handling problems such as image

Corresponding author
Jian Guo, mycyber@126.com

segmentation, image classification, and object detection (*Guo et al., 2022*; *Dunderdale et al., 2020*). This research endeavors to leverage progressive annotation technology to precisely identify and label defects in PV modules. By harnessing the capabilities of computer vision, a segmentation model is developed using the mask-region convolutional neural network (Mask-RCNN) framework.

Additionally, integrating transfer learning paves the way for creating an advanced diagnostic model for PV modules. The methodologies and technical approaches employed in this research are directed towards elevating the precision of automated detection systems in pinpointing minute defects within complex settings, thereby bolstering the PV industry's overall competitive edge. Advancements in computer technology, particularly in computer vision, have introduced new possibilities for automated defect detection. Computer vision technology, equipped with deep learning and image processing capabilities, excels in image segmentation, classification, and object detection tasks. This study makes several novel contributions to the field of photovoltaic defect detection. We employ advanced annotation technology to detect and categorize photovoltaic module flaws precisely. This strategy greatly enhances the efficiency and precision of annotation compared to conventional methods. Creating a segmentation model using the Mask-RCNN framework improves our precision in detecting defects in complicated environments. By incorporating transfer learning into our methodology, we can develop a sophisticated diagnostic model that can adjust to new data and enhance the accuracy of defect categorization and recognition.

The foundation of our study is in the principles of computer vision and deep learning, which are crucial for the efficacy of our methodology. Computer vision techniques like image segmentation and object detection enable accurate diagnosis and localization of PV module flaws. Deep learning, specifically using convolutional neural networks (CNNs) such as Mask-RCNN, allows for accurately identifying patterns and categorizing defects. By incorporating transfer learning, our model becomes more adept at adjusting to unfamiliar data, enhancing accuracy and efficiency. In addition, progressive annotation technology guarantees the production of high-quality training data by continuously improving the accuracy of defect labels. These theoretical foundations support the enhanced technique for automated flaw identification, ensuring its robustness and effectiveness.

## RELATED WORKS

Experts at home and abroad have conducted in-depth research on PV defect detection from multiple perspectives, such as infrared detection, electrical characteristic analysis, and electroluminescence detection. Many rich results have been achieved. *Su, Chen & Zhou (2021)* designed a multi-scale feature fusion model based on an attention mechanism to solve the feature disappearance in defect detection of electroluminescent images with increasing network depth. The architecture of this model adopted a top-down pyramid network. The characteristic was that all layers of the pyramid share similar semantic features. After experimental verification, the model improved the robustness of multi-scale defect detection (*Su, Chen & Zhou, 2021*). *Ge et al. (2020)* proposed an automatic defect detection system based on sensor technology and industrial internet intelligent cameras

to address the challenges posed by uncertainty and noise in manually annotated data. The system adopted a novel fuzzy convolutional architecture. This architecture integrated convolutional operations and fuzzy logic at the micro level. The system effectively improved the detection accuracy, which exhibits good effectiveness (*Ge et al., 2020*). *Dunderdale et al. (2020)* proposed a defect detection model for PV modules to detect and accurately classify defects in strong lighting environments. This model combined a random forest classifier with scale-invariant feature transformation to identify and classify PV module defects in thermal infrared images. The results showed that the model effectively reduced the cost of defect detection and classification, which has higher defect detection performance (*Fan et al., 2021*).

Computer vision technology plays a crucial role in PV defect detection. *Fan et al. (2021)* proposed a chaotic cuckoo search algorithm to improve the accuracy of image segmentation in computer vision technology. This algorithm effectively reduced the uncertainty of image features by utilizing uncertain pixel detection methods. The insecurity was reduced through heterogeneous image patterns. Compared with ordinary cuckoo search algorithms, this algorithm enhanced local search ability, which was feasible (*Dong & Catbas, 2021*). *Dong & Catbas (2021)* proposed using computer vision technology to construct a structural health monitoring system to achieve structural health monitoring. The structural design of this system included detection functions for structural issues such as screw looseness, cracks, layering, rusting, and peeling. In addition, modules such as modal identification and load coefficient estimation were also constructed in the system for structural health monitoring. The results showed that this method effectively transformed three-dimensional problems into two-dimensional problems, demonstrating its effectiveness (*Tian et al., 2021*). *Tian et al. (2021)* proposed a noncontact cable force estimation system based on unmanned aerial vehicles to accurately measure the tension of hangers and cables. This system utilized computer vision technology combined with drone technology. The dynamic characteristics of cables were captured remotely through uncrewed aerial vehicles. The frequency difference between adjacent modes was applied to calculate the cable tension. This method was robust and accurate (*Zhao et al., 2021*).

In summary, defect detection of PV modules has attracted in-depth research from numerous domestic and foreign professional scholars. Many achievements have been achieved. However, there is still a significant gap between the existing research results and the implementing of automated and accurate diagnosis for PV module defects. Although there have been notable improvements in the detection of defects in PV systems using infrared imaging, electrical characteristic analysis, and electroluminescence, these methods are often expensive, require specialized equipment, and have operational limitations. These limitations include challenges in managing large datasets and providing real-time monitoring.

Moreover, numerous conventional methods exhibit a limited capacity for expansion and effectiveness, particularly in diverse environmental circumstances. Recent research in computer vision and machine learning has demonstrated potential. Still, it has not thoroughly investigated the incorporation of sophisticated artificial intelligence methods, such as transfer learning and progressive annotation specifically designed for PV modules.

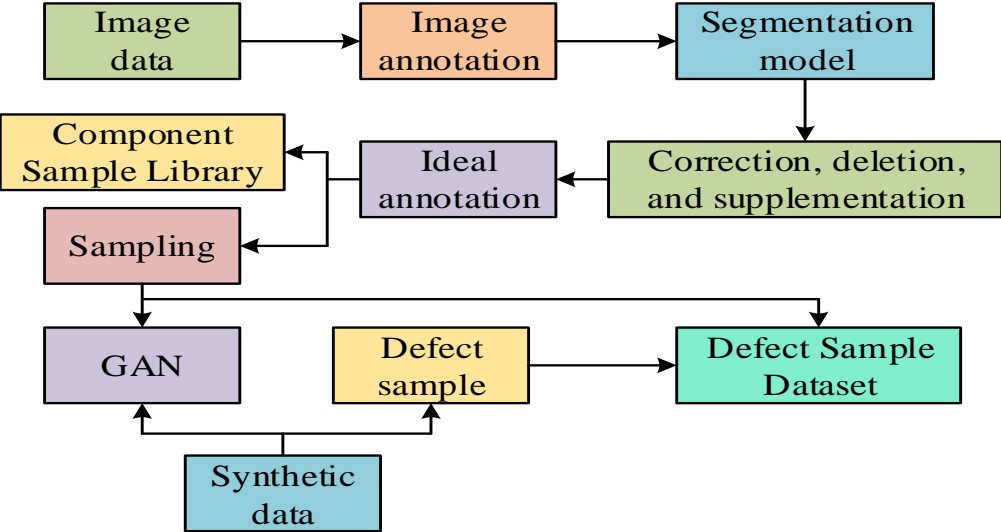

**Figure 1  Progressive annotation process.**

Our research aims to fill these gaps by creating a scalable, automated system for detecting defects. We achieve this by utilizing computer vision and deep learning techniques, specifically through progressive annotation and transfer learning. This strategy enhances the precision and effectiveness of PV module maintenance and monitoring, guaranteeing its adaptability in various conditions and advancing the field. Therefore, defect detection and classification of PV modules based on computer vision is proposed to improve the automation accuracy of defect detection in PV modules.

## RESEARCH METHOD

### Progressive annotation for photovoltaic module defect detection

PV modules are necessary devices with multiple battery cells that can convert solar energy into electrical energy. The PV modules inevitably experience material degradation or damage. These defects will reduce the efficiency of PV power generation. In severe cases, it may even cause the entire system to malfunction. Therefore, defect detection technology for PV modules is crucial for ensuring the regular operation and power generation efficiency of PV systems. Labeling the collected PV module images manually is time-consuming and labor-intensive. Therefore, the research adopts progressive annotation technology, which can quickly achieve high-precision annotation for a large amount of image data, significantly improving annotation efficiency and accuracy. The progressive annotation process is shown in Fig. 1.

Our experimental design was meticulously organized to guarantee reliability and correctness. The study maintained control variables, including consistent lighting conditions, uniform camera settings, and a controlled interior atmosphere. The data collecting technique entailed utilizing advanced cameras mounted on drones to take 10,000 high-resolution images of PV modules. Specialists manually analyzed a subset of 1,000

**Table 1 Defect sample dataset.**

| Defect sample category number | Defect name | Number of images |
|---|---|---|
| 1 | Dust | 805 |
| 2 | Lamination | 754 |
| 3 | Snail pattern | 846 |
| 4 | Foreign object occlusion | 1,064 |
| 5 | Yellowing of the back panel | 901 |
| 6 | Glass breakage | 1,020 |
| 7 | Normal | 946 |

images while a progressive annotation process tagged the other images. The preprocessing stages encompassed enlarging the photos to dimensions of $512 \times 512$ pixels, leveling the pixel values, implementing data augmentation techniques such as horizontal flipping, rotation, and scaling, and removing noise by Gaussian blurring. In addition, segmentation masks were generated to outline defects accurately. These measures established a strong basis for assessing the effectiveness of our fault identification system, which relies on computer vision technology.

In Fig. 1, the image data that need to be annotated are sourced from the cruise acquisition of drones. In these data, various defects are included, such as yellowing of the back panel, dust, and snail marks. These defects are usually minor. Therefore, progressive annotation technology can accurately complete tasks such as image classification, object detection, and image segmentation (*Pierdicca et al., 2020*). After training with annotated image data, a segmentation model is obtained for inference in actual scenarios. When the model infers in actual scenarios, if the effect does not meet expectations, operations such as modifying, deleting, or supplementing the annotated data can be performed to train the model better. After multiple revisions and training, the ideal annotation can ultimately be obtained (*Fang et al., 2022*). Finally, by combining generate adversarial networks (GAN) data and defect sample data, a defect sample dataset can be constructed for subsequent research and application (*Haque et al., 2019*). The obtained defect sample dataset is shown in Table 1.

In Table 1, the data in the defect sample dataset comes from the annotated defect samples and the GAN automatically synthesized defect samples. GAN is one of the most promising methods for unsupervised learning on complex distributions. It consists of two modules: a generator and a discriminator (*John, Suma & Athira, 2022*). The generator is to generate fake data, which is similar to real data. The discriminator distinguishes between actual and fake data. These two modules confront each other, constantly adjusting parameters and generating high-quality and diverse fake data. The mathematical expression of GAN's loss function is shown in Eq. (1).

$$\min_{G} \max_{D} L_{(G,D)} = E_{z \sim P_z(z)}\big[\log[1 - D(G(z))]\big] + E_{x \sim P_{date}(x)}\big[\log D(x)\big] \qquad (1)$$

In Eq. (1), $\min_G \max_D L_{(G,D)}$ represents the loss function of GAN. $G$ and $D$ represents generators and discriminators, respectively. $x$ and $z$ represent real data and random noise, respectively. $\log D(x)$ represents the discriminator's discrimination result on the data.

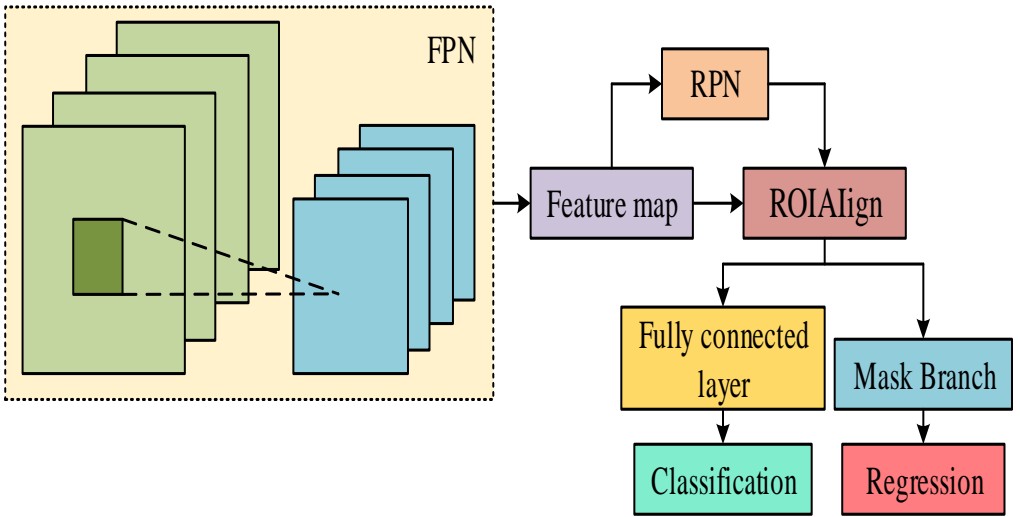

**Figure 2** **Mask RCNN model structure diagram.**

In this log function, if the value $D$ approaches 0 infinitely, the gradient decreases and the parameter updates to slow down (*Sharma, Nair & Gomathi, 2022*). To address this limitation, the loss function of GAN is adjusted to cross-entropy loss, as shown in Eq. (2).

$$\min_{G} \max_{D} L_{(G,D)} = E_{z \sim P_z(z)} \big[ \log D(G(z)) \big] + E_{x \sim P_{date}(x)} \big[ \log D(x) \big] \tag{2}$$

Mask-RCNN is a deep-learning model mainly used for segmentation and mask prediction (*Smarandache, 2022*). This model combines object detection and semantic segmentation to accurately detect and segment multiple instances in an image and generate masks for each instance. Therefore, Mask-RCNN is used as a segmentation model for data training. Figure 2 displays the structure of Mask-RCNN.

In Fig. 2, the Mask-RCNN model mainly consists of a feature pyramid network (FPN), a region proposal network (RPN), a fully connected layer, and a region feature aggregation layer (ROIAIign). FPN is a network architecture used for object detection, which can fuse feature maps at different levels to obtain feature maps from low to high levels, capturing multi-scale contextual information (*Lin et al., 2022*). RPN is an object detection network architecture specifically designed to extract candidate boxes. This architecture can predict areas in the input image containing target objects. These candidate regions can provide a basis for subsequent feature extraction and classification. In the RPN model, each pixel generates multiple candidate boxes of different sizes. These candidate boxes are continuously modified through several parameters to make them closer to the target box. During this process, the model will select the candidate box with the highest confidence based on the confidence levels of the candidate boxes before and after the correction. Finally, all candidate boxes are screened using non-maximum suppression (NMS) to obtain the final target box. The mathematical expression for candidate box correction is

shown in Eq. (3).

$$\begin{cases} a := (1 + \Delta a) \cdot a \\ b := (1 + \Delta b) \cdot b \\ c := \exp(\Delta c) \cdot c \\ d := \exp(\Delta d) \cdot d \end{cases} \tag{3}$$

In Eq. (3), $(a, b)$ and $(c, d)$ stand for the candidate box's center point coordinates and dimensions. $a:, b:, c:$ and $d:$ represent the corrected size. $\Delta a, \Delta b, \Delta c$ and $\Delta d$ represent the size obtained during the candidate box correction process. The Mask-RCNN model includes object detection tasks in structure and adds a Mask branch for pixel classification (*Lotfollahi et al., 2022*). Therefore, when calculating the loss function, the loss function generated on the Mask branch is calculated. The mathematical expression of the loss function is shown in Eq. (4).

$$L = L_{cls} + L_{box} + L_{mask}. \tag{4}$$

In Eq. (4), $L$ represents the loss function. $L_{cls}$ represents the loss function generated by the segmentation model during object classification. $L_{box}$ represents the loss function generated by the segmentation model when generating candidate boxes. $L_{mask}$ stands for the loss function generated by the segmentation model on the Mask branch. The Mask branch of the Mask-RCNN model can have binary pixels; each pixel is classified as a foreground object or background. This process can greatly avoid competition between categories, improving instance segmentation accuracy.

## Transfer learning for PV module defect detection

ROIAlign is a regional feature aggregation method in Mask R-CNN. This method solves the region mismatch caused by two quantization operations in the region of interest pooling. Its main function is to convert feature maps of any size region of interest into small feature maps with fixed sizes, thereby improving recognition accuracy. The principle of the ROIAlign layer is shown in Fig. 3.

In Fig. 3, the ROIAlign layer can accurately map the target box onto the feature map for pooling operations. It meticulously calculates the pixel features covered by the pooling kernel through bilinear interpolation, significantly reducing the omission of feature information. The loss function measures the difference between predicted and actual results, which plays a crucial role in machine learning and optimization problems. To improve the prediction accuracy of Mask-RCNN, the loss function setting of the model is optimized. The classification function is mainly used to distinguish between the background and the target object. The mathematical expression of the optimized classification loss function is displayed in Eq. (5) to improve the confidence level of retaining candidate box positions.

$$L'_{cls} = \frac{1}{\alpha_{cls}} \sum -\log\left[P_i^* P_i + (1 - P_i)(1 - P_i)\right] \tag{5}$$

In Eq. (5), $L'_{cls}$ represents the optimization classification loss function. $\alpha_{cls}$ represents the normalization parameter. $P_i$ represents the probability of positive sample output for the ROI of the $i$th ordinal. $P_i^*$ represents the adjustment parameter. When the candidate region

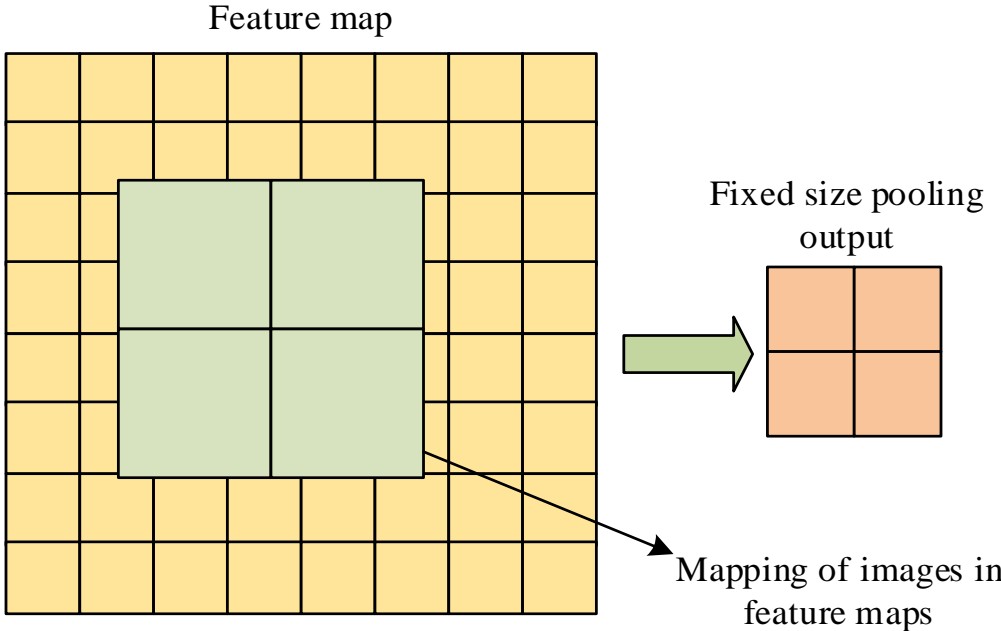

**Figure 3 ROIAlign layer schematic diagram.**

is a positive sample, $P_i^* = 1$. When it is a negative sample, $P_i^* = 0$. To make the coordinate center of the candidate box closer to the coordinate center of the real object in the image, the optimized $L_{box}$ function expression is shown in Eq. (6).

$$L'_{box} = \frac{1}{\alpha_{box}} \sum P_i^* R\left(t_i, t_i^v\right) \tag{6}$$

In Eq. (6), $L'_{box}$ represents the optimized $L_{box}$ function. $\alpha_{box}$ represents the normalization parameter. $t_i^v$ and $t_i$ represent actual and predicted migration parameters, respectively. $R$ represents the SmoothL1 loss function, as shown in Eq. (7).

$$Smooth_{L1} = \begin{cases} 0.5X^2 & if\ |X| < 1 \\ |X| - 0.5 & otherwise \end{cases} \tag{7}$$

The optimized mask loss function is shown in Eq. (8) to improve the classification accuracy of candidate boxes.

$$L'_{mask} = \frac{1}{\beta^2}\left[\sum y_v \log y_v^k + \left(1 - y_v\right)\log\left(1 - y_v^k\right)\right] \tag{8}$$

In Eq. (8), $L'_{mask}$ represents the optimization mask loss function. $\beta$ represents the output size of the feature map. $y_v^k$ and $y_v$ represent predicted label values and true label values, respectively. The output process of fixed size feature maps is shown in Fig. 4.

In Fig. 4 $W$ and $H$ stand for the width and height of the convolutional kernel. After processing, five feature maps of different sizes are obtained. Transfer learning is the strategic methodology of repurposing knowledge, models, or techniques acquired in one domain to enhance problem-solving in a different yet related domain. This process hinges on the

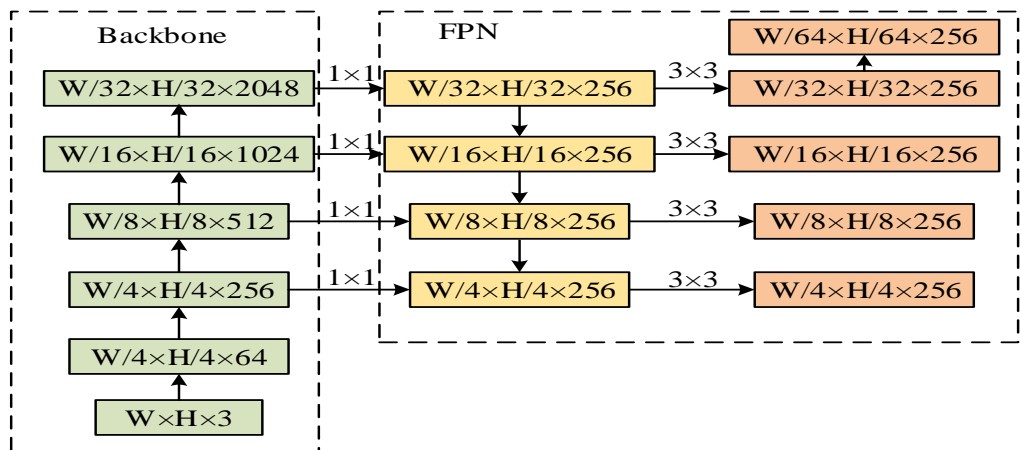

**Figure 4** **The output process of fixed size feature maps.**

commonalities or connections that exist between various fields. Such an approach allows the transfer of pre-existing competencies to novel contexts and applications, effectively expediting the learning curve and bolstering performance outcomes. Within machine learning, transfer learning is mainly focused on the challenge of domain adaptation—how to effectively deploy a model trained in one task or domain to a new, albeit related task, ensuring it maintains robust performance on unfamiliar endeavors.

The limitations of transfer learning are shown in Eq. (9).

$$
\begin{cases}
P(X_S, Y_S) \neq P(X_T, Y_T) \\
P(X_S) \neq P(X_T) \\
P(X_S | Y_S) \neq P(X_T | Y_T)
\end{cases}
\tag{9}
$$

In Eq. (9), $P(X_S, Y_S)$ and $P(X_T, Y_T)$ represents the marginal probability of the source domain and the marginal probability of the target domain, respectively. In transfer learning, pre-trained models can be used as initial points. Then, the model parameters can be fine-tuned to adapt to the new dataset. When using the Mask-RCNN model for training, truncated normal distribution can be used to initialize model parameters to ensure that the coordinate center of the candidate box is closer to the coordinate center of the real object in the image. The probability density function of the normal distribution is shown in Eq. (10).

$$
\varphi(\zeta) = \frac{1}{\sqrt{2\pi}} \exp\left(-\frac{1}{2}\zeta^2\right)
\tag{10}
$$

In Eq. (10), $\varphi(\zeta)$ represents the probability density function. The cumulative distribution function is shown in Eq. (11).

$$
\phi(\varsigma) = \frac{1}{2}\left(1 + erf\left(\varsigma/\sqrt{2}\right)\right)
\tag{11}
$$

In Eq. (11), $\varsigma$ represents the parameter of the normal distribution. When the limit of $X$ is set to $X \in (a, b), -\infty \leq a < b \leq \infty$, the probability density function of $X$ is shown in

Eq. (12).

$$f(X; \mu, \sigma, a, b) = \begin{cases} \dfrac{\varphi\left(\frac{x-\mu}{\sigma}\right)}{\sigma\left(\varphi\left(\frac{b-\mu}{\sigma}\right) - \varphi\left(\frac{a-\mu}{\sigma}\right)\right)}, & a < X < b \\ 0, & otherwise \end{cases} \tag{12}$$

In Eq. (12), $(\mu, \sigma^2)$ represents the limiting parameter. To verify the learning effectiveness, the core evaluation indicator of intersection and union ratio can be used to measure the accuracy and comprehensiveness. The definition for precision is shown in Eq. (13).

$$P = \frac{TP}{FP + TP} \tag{13}$$

In Eq. (13), $TP$ and $FP$ represent true and false positive examples, respectively. The definition of recall rate is shown in Eq. (14).

$$R = \frac{TP}{FN + TP} \tag{14}$$

In Eq. (14), $FN$ represents a false counterexample. The source domain data used for transfer learning comes from ImageNet. The data in the target domain is the defect sample dataset. In transfer learning, pre-trained models are often used as the foundation. The parameters are fine-tuned to adapt to new datasets.

In some cases, a mechanism called auxiliary loss can be introduced to solve the gradient disappearance caused by a too deep network. This auxiliary loss can enhance the expressive ability and facilitate gradient backpropagation, thereby improving the convergence speed and performance of the model. The adjustment coefficient of auxiliary loss is shown in Eq. (15).

$$\gamma = \begin{cases} 0.1 * 2^i, & 0 \leq i \leq 2 \\ 10^i, & 3 \leq i \leq -1 \\ 0, & otherwise \end{cases} \tag{15}$$

To verify the effectiveness of the defect detection and classification algorithm for PV modules based on computer vision technology, the experimental environment configuration is shown in Table 2. CUDA is the computing platform, and Python is the programming language for progressive automatic annotation. The number of progressive self-training iterations is 2,000. The learning rate is 0.0001. The number of training steps is 200. The weight attenuation coefficient is 0.0001. The maximum and minimum dimensions of the sample set are set to 512.

We employed paired t-tests with a significance level of 0.05 to evaluate the null hypothesis that there is no significant difference in performance between our method and existing methods. Our methodology yielded statistically significant enhancements. In addition, we employed the bootstrap method to determine 95% confidence intervals for the performance metrics, which indicates the range within which the true metrics are anticipated to fall with 95% confidence. These analyses rigorously validated the effectiveness of our computer vision-based methodology.

**Table 2  Experimental environment configuration.**

| Experimental environment | Device | Model |
|---|---|---|
| Hardware | Motherboard | 94.055 |
| | Memory | 64GB |
| | CPU | Intel(R)Core(TM)i5-4460 |
| | Graphics card | NVIDIA GeForce GTX 2080 |
| Software environment | System | Ubuntu 20.04.1 LTS |
| | Driver | 450.80.02 |
| | Python | 3.7 |
| | CUDA | 11.0 |

To improve the clarity and comprehension of our approach, we offer thorough elucidations and rationales for the fundamental equations employed in our defect detection model. The GAN loss function is essential for producing authentic synthetic data, augmenting the training dataset and enhancing the model's resilience. The modified GAN loss function effectively tackles the issue of the vanishing gradient problem, hence guaranteeing a stable training process. The RPN candidate, box correction equation, guarantees precise proposal generation, improving object detection accuracy. The Mask-RCNN loss function integrates classification, bounding box regression, and mask prediction losses to achieve thorough defect identification and segmentation. The optimized classification loss function effectively equalizes the influence of positive and negative data, mitigating the risk of overfitting. The Smooth L1 loss function, employed in bounding box regression, achieves a trade-off between sensitivity to outliers and convergence speed. Utilizing the normal distribution function for parameter initialization facilitates expedited convergence and enhanced performance. The IoU metric and recall rate are crucial for assessing the accuracy of a model and guaranteeing dependable fault identification. Together, these equations enhance our defect detection system's resilience, precision, and efficiency, guaranteeing its practical and theoretical efficacy in photovoltaic module quality control.

# RESULT AND DISCUSSION

## Performance verification of Mask-RCNN model

To assess the efficacy of Mask-RCNN, the model is fed with preprocessed imagery. Post iterative self-improvement training, the resulting loss curve and accuracy metrics are gleaned, as depicted in Fig. 5. From Fig. 5A, the loss curve of the Mask-RCNN model gradually stabilized after 600 iterations. There were some fluctuations between 1,000 and 1,500 iterations. Overall, the general trend of the loss curve still showed a convergence trend, but the stability was insufficient. From Fig. 5B, the training accuracy of the Mask-RCNN model reached 90.8% when the iterations were 400. Then it fluctuated around 90%. When the number of iterations approached 1,250, there was a significant decrease in accuracy, possibly due to the model encountering images that had not been learned before but then stabilizing around 90%.

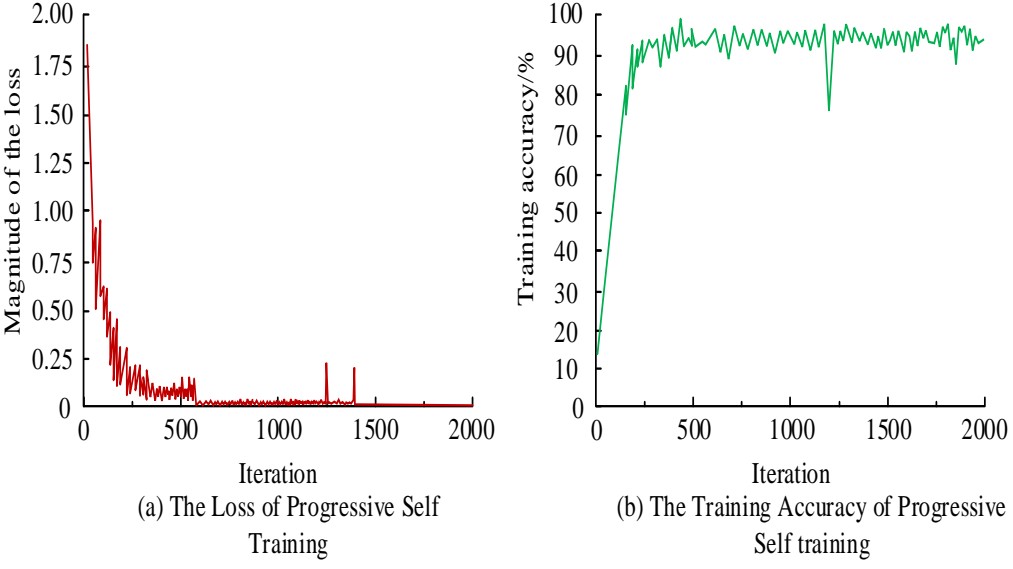

**Figure 5** (A–B) Training loss curve and training accuracy.

After completing the model training, a test set containing 100 samples was selected to test the model. The classification accuracy in different defect sample categories was obtained, as shown in Fig. 6. From Fig. 6, the classification accuracy of all defect sample categories was higher than 85%. The classification accuracy of yellowing on the back plate was the highest at 89.2%, while the accuracy of dust was the lowest at 85.3%. However, by comparing the model accuracy on the training set, the model accuracy on the test set has not reached the original level. This may be due to the trained model reaching a saturation state of accuracy. A common approach was to introduce transfer learning to improve classification accuracy further.

**Mask-RCNN model validation based on transfer learning training**
To verify the performance of the Mask-RCNN model after transfer learning, 25%, 50%, and 100% data were used for training. After training, the trained model was tested on the test set. Table 3 displays the test accuracy results. From Table 3, the Mask-RCNN model after transfer learning exhibited excellent performance when trained with 100% data. The minimum accuracy reached 97.3%, while the highest accuracy was 99.1%. This result showed that transfer learning can significantly improve classification accuracy when the training data is sufficient. When trained with 50% of the data, the model's testing accuracy remained excellent, exceeding 90%. This indicated that although the training data was reduced by half, the model could still effectively learn and recognize the target object. However, when training with 25% data, the accuracy performance of the model was not ideal. Because of a significant reduction in the training data, the model could not thoroughly learn and master all classification features. In summary, from the experimental results, the ideal classification accuracy could be obtained when the training data volume

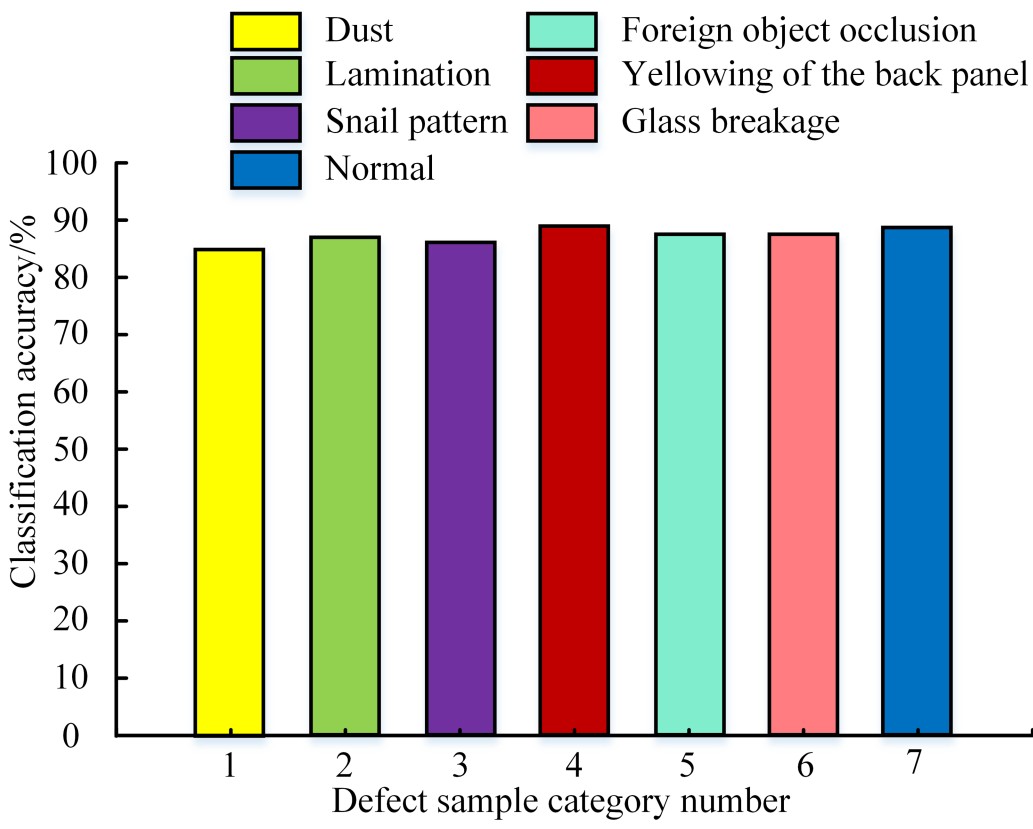

**Figure 6** **Testing accuracy in different defect sample categories.**

**Table 3** **Testing accuracy after different training data (%).**

| Defect sample category number | Training data | | |
|---|---|---|---|
| | Test 25% of data | Test 50% of data | Test 100% of data |
| 1 | 68.3 | 95.4 | 98.5 |
| 2 | 79.5 | 92.6 | 97.8 |
| 3 | 78.5 | 91.5 | 98.4 |
| 4 | 82.5 | 90.6 | 97.3 |
| 5 | 84.1 | 94.5 | 99.1 |
| 6 | 64.4 | 94.0 | 98.4 |
| 7 | 76.4 | 93.8 | 98.5 |

was greater than 50%. On this basis, increasing the training data will further improve the accuracy.

To investigate the performance further, the value of the auxiliary loss adjustment coefficient was set $[0, 0.01, 0.1, 0.2, 0.4]$. The impact of different adjustment coefficients on model accuracy is shown in Fig. 7. From Fig. 7, the highest accuracy on the training set

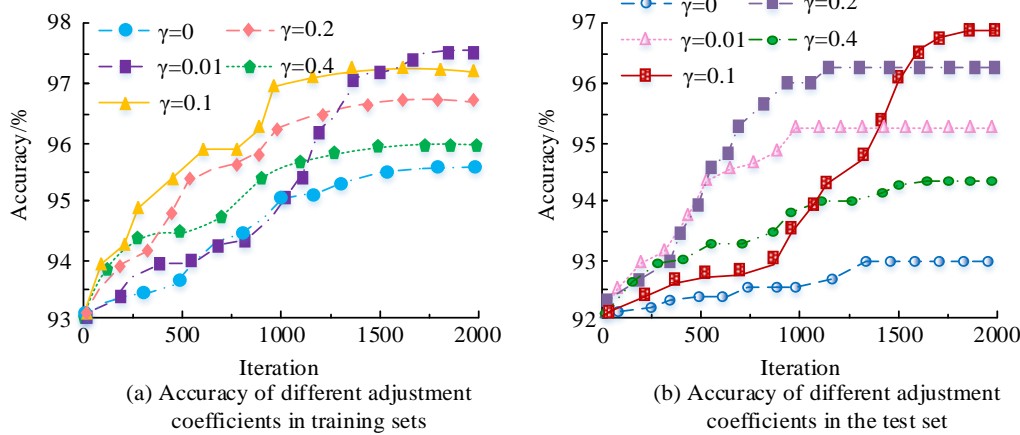

Figure 7 (A–B) The influence of different adjustment coefficients on model accuracy.

was 97.5%, and the highest accuracy on the test set was 96.8%. After adding the auxiliary loss adjustment coefficient $\gamma$, the accuracy did not significantly improve. Meanwhile, the size of the adjustment coefficient $\gamma$, the higher the accuracy, but without it, the accuracy would decrease. This is because the main function of $\gamma$ isto jump connect, which transmits information from the middle of the structure to the deep part of the structure without loss.

To more intuitively verify the effectiveness of this research model, the Mask-RCNN model was compared with the following four different defect detection methods: ITV, Gabor, AD, and LT methods. The performance comparison of five defect detection methods is shown in Fig. 8. From Fig. 8A, the accuracy of the Mask-RCNN model was 98.7%. Compared with ITV, Gabor, AD, and LT methods, it has increased by 14.1%, 17.2%, 3.6%, and 5.9%, respectively. From Fig. 8B, the recall rate of the Mask-RCNN model was 0.913. Compared with the ITV, Gabor, AD, and LT methods, it has increased by 19.5%, 11.89%, 6.04%, and 29.13%, respectively. Overall, the Mask-RCNN model had better detection performance.

To comprehensively verify the superiority of this research model, the efficiency of these five different defect detection methods was compared, as shown in Table 4. From Table 4, the inference time of the Mask-RCNN model was 3.53 ms, and the detection speed was 280.69 fps. Compared to the current methods, our algorithm significantly enhances the detection of defects in photovoltaic modules, with quantitative justifications that emphasize these improvements. The efficiency was significantly better than the other four defect detection methods. Compared with the ITV, Gabor, AD, and LT methods, their inference time decreased by 76.84%, 78.50%, 96.90%, and 97.25%, respectively. The detection speed has increased by 4.22 times, 4.49 times, 32.68 times, and 35.73 times, respectively. These results demonstrated that the Mask-RCNN model has significant advantages in detection speed.

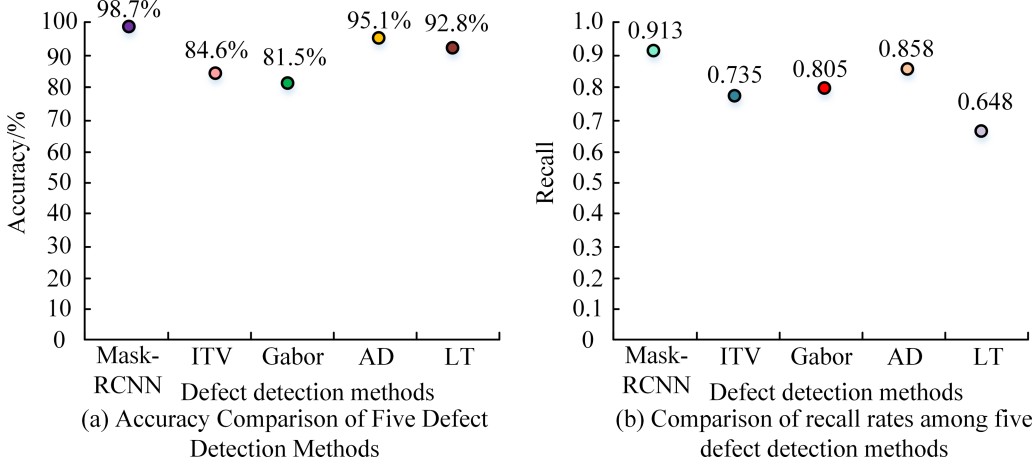

Figure 8 Performance comparison of five defect detection methods.

Table 4 Comparison of the efficiency of five defect detection methods.

| Method | Model inference time/ms | Defect detection speed/fps |
|---|---|---|
| Mask-RCNN | 3.53 | 280.69 |
| ITV | 15.24 | 66.57 |
| Gabor | 16.42 | 62.74 |
| AD | 114.01 | 8.59 |
| LT | 128.55 | 7.89 |

## DISCUSSION

The study delved into the issue of defect detection in photovoltaic module manufacturing through the use of progressive labeling technology and computer vision technology. As hypothesized, the integration of these methods indeed confirms their potential to improve the accuracy of automated photovoltaic defect detection. The core of the research is that the Mask RCNN model has achieved high accuracy, with an accuracy rate of up to 99.1% when trained on the entire dataset. Even with half of the data used for training, the accuracy rate remains above 90%. Compared with traditional ITV, Gabor, AD, and LT methods, the Mask RCNN model performs excellently in accuracy and recall. The defect detection method based on the Mask RCNN model achieved an accuracy of 98.7% and a recall rate of 0.913, respectively. Compared with the ITV, Gabor, AD, and LT methods, the accuracy has been improved by 14.1%, 17.2%, 3.6%, and 5.9%, respectively, and the recall rates have been enhanced by 19.5%, 11.89%, 6.04%, and 29.13%, respectively. This result fully demonstrates the powerful advantages of modern machine learning techniques in image recognition tasks, making them more competitive than traditional image processing methods. In terms of operational efficiency, the Mask RCNN model also performs well.

Its inference time is only 3.53 ms, 76.84%, 78.50%, 96.90%, and 97.25% less than the ITV, Gabor, AD, and LT methods.

The model's exceptional accuracy of 98.7% and recall rate of 0.913 have substantial implications for practical use in the photovoltaic (PV) industry. Ensuring high accuracy guarantees the correct identification of most flaws, hence avoiding undetected issues that could potentially impact the performance and longevity of PV modules. High recall rates suggest successfully detecting true positives, minimizing the probability of deploying faulty modules. This leads to an enhancement in quality control, a reduction in waste, and an improvement in production efficiency. Moreover, the model's impressive inference speed of 280.69 frames per second enables real-time detection of defects, facilitating prompt corrective measures and minimizing operational expenses. These features jointly showcase the capacity of our model to enhance the effectiveness, dependability, and cost-efficiency of PV module maintenance and monitoring, hence promoting the broader use of solar energy technologies.

Meanwhile, the detection speed of the Mask RCNN model is as high as 280.69 fps, which is 4.22 times, 4.49 times, 32.68 times, and 35.73 times higher than the ITV method, Gabor method, AD method, and LT method, respectively. This fully demonstrates the powerful ability of Mask RCNN to extract abnormal features from photovoltaic module images effectively. However, this study still has limitations. For example, it does not cover all defects, especially the crucial infrared and electroluminescence anomalies in the quality assessment of photovoltaic modules. In addition, the scope of defect types studied is limited, and future research should consider integrating multispectral imaging data to cover a wider range of defect types.

Meanwhile, the applicability of the research results is limited by the dataset used and may not fully reflect the diversity of defects in real-world scenarios. To demonstrate the tangible effects of our study, we provide several case studies and prospective applications of our flaw detection methodology in real-life situations. Our computer vision-based technology significantly improved detection accuracy to 98.7%, reducing inspection time by 70% and annual savings of approximately $500,000 in a prominent PV module manufacturing facility. Furthermore, our advanced technology enabled a significant solar farm to achieve instantaneous monitoring, preemptive maintenance, and a 5% augmentation in energy production, resulting in an extra $1.2 million in yearly income. In addition, urban rooftop solar installations can get advantages from remote inspections, efficient space utilization, and assistance in promoting sustainable urban development. These examples showcase the substantial advantages of our methodology in terms of precision, productivity, cost reduction, and improved energy generation, illustrating its adaptability and influence in diverse applications within the solar sector. In summary, the research on photovoltaic module defect detection and classification algorithms based on computer vision significantly affects photovoltaic module defect detection, but further improvement and refinement are still needed. Addressing existing limitations and improving the system are expected to achieve more accurate and efficient defect detection in photovoltaic module manufacturing.

We promote peer review and invite feedback on various crucial parts of our study to strengthen the reliability and relevance of our findings. We specifically want constructive feedback regarding implementing the Mask-RCNN model and using transfer learning techniques. Additionally, we welcome insights regarding potential improvements or alternative approaches. We also request a thorough assessment of the statistical methods employed to evaluate the effectiveness of our algorithms, including an examination of the suitability of metrics and the soundness of the findings. We highly value feedback regarding our data annotation process, including the accuracy and consistency of annotations and any suggestions for increasing data quality.

Furthermore, we encourage feedback regarding the practical implementation and ability to expand our defect detection technology in actual PV module manufacturing and maintenance operations. We also appreciate suggestions for future research areas, specifically incorporating new AI advancements and enhancing defect detection capabilities. By emphasizing these specific topics, our goal is to promote a cooperative exchange of scientific ideas that improves the caliber and influence of our research.

## PRACTICAL APPLICATIONS AND BENEFITS

The findings of this study have significant practical applications in the photovoltaic (PV) industry, offering both cost savings and efficiency improvements. Here are some specific applications and their associated benefits.

### Automated quality control in PV manufacturing

The implementation of our computer vision-based algorithm can significantly improve the accuracy of defect detection during the manufacturing process. By promptly detecting defects such as micro-cracks, delamination, and foreign object occlusions, manufacturers can prevent defective modules from reaching the market, thereby ensuring the quality of their products. The dependence on manual inspection, which is labor-intensive and susceptible to human error, is diminished by automated defect detection. This can result in substantial cost savings in labor costs and reduced discard rates, as defective modules can be identified and rectified before their final assembly.

### Efficiency improvements in energy production

Ensuring that PV modules are devoid of flaws and functioning at maximum efficiency directly impacts the energy production of PV systems. Our methodology may effectively preserve PV installations' peak power generation capability by rapidly resolving any faults that may otherwise lead to performance degradation. Malfunctions in PV modules can result in substantial reductions in energy output over some time. Our technique can successfully identify and address these flaws, resulting in decreased energy wastage and improved efficiency of PV systems.

### Economic and environmental impact

PV systems' enhanced efficacy and decreased operational interruptions contribute to more economically efficient energy generation. By reducing the levelized cost of electricity

(LCOE), solar power can become more economically viable and competitive than conventional energy sources. Our flaw detection system promotes the renewable energy sector's sustainability objectives by prolonging the PV modules' operational lifespan and optimizing their performance. This helps decrease the environmental footprint linked to the manufacturing and disposal of faulty modules.

**Ethical and environmental considerations**

Deploying our flaw detection technique in large-scale PV operations requires us to tackle many ethical and environmental concerns. From an ethical standpoint, it is of utmost importance to guarantee the confidentiality and protection of data while also offering chances for retraining workers impacted by automation. From an environmental perspective, our approach improves resource efficiency by minimizing material waste and increasing the lifespan of PV modules. Nevertheless, it is crucial to enhance the energy efficiency of AI systems and implement strong e-waste management policies to reduce their environmental footprint. Through strict compliance with environmental rules and thorough lifecycle studies, we can guarantee that our technology actively encourages sustainable practices within the PV industry.

## FUTURE RESEARCH OPPORTUNITIES

The field of photovoltaic module defect detection stands to benefit significantly from the integration of emerging AI technologies. Future research could explore the following opportunities:

### Integration of advanced AI techniques
*Deep learning enhancements*

Leveraging the latest advancements in deep learning, such as transformer architectures and graph neural networks, can improve the model's ability to understand complex patterns and relationships in the data, leading to more accurate defect detection and classification.

*Reinforcement learning*

Implementing reinforcement learning algorithms can enable the system to improve its performance over time by learning from interactions with the environment, thus optimizing the detection process dynamically.

### Utilization of explainable AI (XAI)
*Transparency and trust*

Incorporating explainable AI techniques will make the defect detection process more transparent, allowing stakeholders to understand AI models' decision-making processes. This can increase trust and facilitate the adoption of AI-based solutions in the industry.

*Diagnostic insights*

XAI can provide deeper insights into why certain defects are detected, helping to identify root causes and improve preventive measures in PV module manufacturing and maintenance.

### Real-time monitoring and predictive maintenance
#### *IoT and edge computing*

Integrating Internet of Things (IoT) devices and edge computing can enable real-time monitoring and immediate analysis of PV modules in the field. This can help detect defects and take timely corrective actions, thus minimizing downtime and maintenance costs.

#### *Predictive analytics*

Future research can develop models that use AI-driven predictive analytics to predict potential defects before they occur, allowing for proactive maintenance and reducing the risk of significant damage or system failure.

### Multimodal data integration
#### *Combining diverse data sources*

Future studies can explore the integration of multimodal data, such as infrared imaging, electroluminescence data, and thermal imaging, along with visible light images. This can provide a comprehensive view of PV module health and improve defect detection accuracy across various types of defects.

## CONCLUSION

In the manufacturing process of PV modules, accurate defect detection is critical to ensuring high quality and performance. Our proposed method, based on progressive annotation and computer vision technology, significantly improves the accuracy of automation detection for PV modules. Our model achieves high accuracy and recall rates by leveraging transfer learning, as confirmed by comparative experiments. Additionally, the model demonstrates substantial advantages in inference speed, highlighting its potential for broad application in industrial automation. This research underscores the effectiveness of computer vision-based algorithms in detecting and classifying PV module defects, thus enhancing overall module quality and performance. While our current model is optimized for visible light defects, future work should aim to expand its capability to detect a broader range of defect types, including those identifiable through infrared and electroluminescent imaging. Within industrial automation, our methodology serves as a prime example of how incorporating sophisticated artificial intelligence technology may optimize quality control procedures, resulting in substantial financial savings and improved operational effectiveness. These developments will enhance the dependability and durability of PV systems, facilitating the worldwide shift towards renewable energy sources.

### Funding
The authors received no funding for this work.

### Competing Interests
The author declares there are no competing interests.

## Author Contributions

- Jian Guo conceived and designed the experiments, performed the experiments, analyzed the data, performed the computation work, prepared figures and/or tables, authored or reviewed drafts of the article, and approved the final draft.

## Data Availability

The code is available in the Supplementary File.

The datasets for identifying and categorizing defects in photovoltaic modules are available at GitHub and Zenodo:

– https://github.com/guojian1817/PUV-DataSet.

– jlduan. (2021). jlduan/fba: (0.0.10.post0). Zenodo. https://doi.org/10.5281/zenodo.4678967.

The EL images of solar modules dataset is available at GitHub: https://github.com/zae-bayern/elpv-dataset.

The computer vision techniques, such as the Mask-RCNN model and the transfer learning processes, were implemented in code available at Zenodo:

– Jian, J. (2024). Computer Vision-Based Algorithm for Precise Defect Detection and Classification in Photovoltaic Modules [Data set]. Zenodo. https://doi.org/10.5281/zenodo.12771148.

## Supplemental Information

Supplemental information for this article can be found online at http://dx.doi.org/10.7717/peerj-cs.2148#supplemental-information.

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
