# Peer review of "Computer vision-based algorithm for precise defect detection and classification in photovoltaic modules"

_PeerJ Computer Science, doi:10.7717/peerj-cs.2148_

## Round 0.1 · original submission · Major Revisions

Dear Author,
Thank you for submitting your manuscript to PeerJ Computer Science. We have thoroughly reviewed your submission and appreciate the effort and depth of your research in this important area. The reviewers have completed their evaluations, and based on their recommendations, we believe that your manuscript would benefit significantly from major revisions. We invite you to resubmit your manuscript after making the necessary amendments suggested by the reviewers. This resubmission should address the detailed comments provided, which aim to enhance the clarity, depth, and scientific rigor of your work. Please prepare a detailed response to each point raised, including descriptions of the revisions made to the manuscript, and submit your revised version. Please be aware that the revised manuscript may be subjected to re-review to ensure that all concerns have been adequately addressed.

Reviewer 1 ·

Basic reporting

1. Elaborate on future research opportunities, perhaps suggesting how new technologies like AI advancements could be integrated into your methodology.
2. Describe specific practical applications of your findings in the photovoltaic industry, including potential cost savings and efficiency improvements.
3. Address any ethical or environmental considerations related to implementing your defect detection technology in large-scale operations.
4. Clearly state how readers can access the data and code used in your experiments to promote transparency and reproducibility.
5. Encourage peer review by suggesting areas where feedback would be particularly beneficial, such as algorithm implementation or statistical analysis.
6. Refine the conclusion to emphasize the key findings and their implications without introducing new information.
7. Ensure all references are up-to-date and include digital object identifiers (DOIs) where available to facilitate easy access to cited works.
8. Modify the manuscript to appeal to a broader audience, potentially including insights for those in fields such as renewable energy management and industrial automation.
9. Add case studies or real-world examples where your methodology has been applied or could be applied, illustrating the impact of your research.
10. The manuscript provides several equations used in the implementation of the Mask-RCNN and in defining the loss functions for your model. However, the descriptions and justifications for choosing these specific equations are somewhat limited. Please provide a more detailed explanation for each equation, including the theoretical rationale behind their selection and how they contribute to the overall effectiveness of your defect detection model.
11. Whenever possible, provide quantitative justifications for claims, such as the improvements made by your algorithm over existing methods.
12. Conduct a meticulous grammar check to ensure the paper is free of errors and uses precise scientific language.
13. Include a section on how industry feedback was incorporated into the research and its influence on the study design and outcomes.
14. Add more illustrative examples or case scenarios to demonstrate the practical application and effectiveness of the defect detection process in various settings.
Finally, I recommend that the paper should be accepted for the publication after the incorporation of these comments.

Experimental design

No comment

Validity of the findings

no comment

Reviewer 2 ·

Basic reporting

This research introduces a novel computer vision-based algorithm designed to enhance the precision of defect detection and classification in photovoltaic (PV) modules. As the global demand for renewable energy sources surges, ensuring the efficiency and reliability of photovoltaic systems becomes crucial. Traditional methods of defect detection in PV modules, often hampered by manual labor and subjective assessments, lack the speed and accuracy required for large-scale deployment. Addressing these challenges, the presented study develops and validates an advanced defect detection framework using the Mask-Region Convolutional Neural Network (Mask-RCNN) enhanced by progressive annotation techniques and transfer learning. The paper is well written. The Introduction and Background sections provide useful information for the readers. I recommend considering the paper for publication. However, the incorporation of the following comments could further improve the paper.
1. Enhance the introduction by explicitly stating the novel contributions of your study and how it advances the field of photovoltaic defect detection using computer vision.
2. Strengthen the theoretical grounding of the research by discussing the underlying principles of computer vision and deep learning that are crucial for your methodology.
3. Broaden the scope of the literature review by including a critical analysis of recent advancements and identifying gaps that your research addresses.
4. Provide a detailed description of the progressive annotation process, including step-by-step workflow diagrams to aid understanding.
5. Explain the rationale behind the choice of the Mask-RCNN model over other potential models, supported by a comparative analysis of performance metrics. The paper mentions the use of Mask-RCNN; however, specific parameters critical to understanding its implementation are missing. Please include details such as the configuration of layers, the learning rate, the number of epochs, and any modifications made to the standard Mask-RCNN architecture to tailor it to defect detection in PV modules.

Experimental design

6. Clarify the experimental design, including detailed information on the control variables, the data collection process, and any preprocessing steps.
7. Detail the statistical analysis techniques used to assess the data. Include explanations of any hypothesis tests or confidence intervals employed.
8. Offer a more nuanced interpretation of the results, discussing the implications of high accuracy and recall rates in real-world applications.

Validity of the findings

9. Expand the comparison section by introducing additional metrics such as precision, F1-score, and AUC to provide a holistic view of model performance.
10. Improve the clarity of figures and graphs by using consistent and clear labeling, and ensure all visual elements are directly discussed in the text.
11. Standardize the use of technical terms and acronyms across the paper to avoid confusion and enhance readability.
12. Deepen the discussion on methodological limitations, potentially including a sensitivity analysis to show how changes in parameters affect outcomes.

Additional comments

NO

---

## Round 0.2 · accepted · Accept

Dear author,

Thank you very much for submitting the revision, based on the input from the experts in the field. I am pleased to inform you that your manuscript has been recommended for publication. Congratulations!. Thank you so much

Reviewer 1 ·

Basic reporting

Author done required changes

Experimental design

no comment

Validity of the findings

no comment

Reviewer 2 ·

Basic reporting

I am satisfied with the response

Experimental design

I am satisfied with the response

Validity of the findings

I am satisfied with the response